# Effects of Baicalein and Chrysin on the Structure and Functional Properties of β-Lactoglobulin

**DOI:** 10.3390/foods11020165

**Published:** 2022-01-09

**Authors:** Ang Li, Lei Chen, Weijie Zhou, Junhui Pan, Deming Gong, Guowen Zhang

**Affiliations:** State Key Laboratory of Food Science and Technology, Nanchang University, Nanchang 330047, China; arf1023068736@163.com (A.L.); lchen508@163.com (L.C.); zhouwj9951997@163.com (W.Z.); panjunhui@ncu.edu.cn (J.P.); dgong01@gmail.com (D.G.)

**Keywords:** baicalein, chrysin, β-lactoglobulin, functional property, spectrophotometry, molecular dynamics simulation

## Abstract

Two flavonoids with similar structures, baicalein (Bai) and chrysin (Chr), were selected to investigate the interactions with β-lactoglobulin (BLG) and the influences on the structure and functional properties of BLG by multispectral methods combined with molecular docking and dynamic (MD) simulation techniques. The results of fluorescence quenching suggested that both Bai and Chr interacted with BLG to form complexes with the binding constant of the magnitude of 10^5^ L·mol^−1^. The binding affinity between BLG and Bai was stronger than that of Chr due to more hydrogen bond formation in Bai–BLG binding. The existence of Bai or Chr induced a looser conformation of BLG, but Chr had a greater effect on the secondary structure of BLG. The surface hydrophobicity and free sulfhydryl group content of BLG lessened due to the presence of the two flavonoids. Molecular docking was performed at the binding site of Bai or Chr located in the surface of BLG, and hydrophobic interaction and hydrogen bond actuated the formation of the Bai/Chr–BLG complex. Molecular dynamics simulation verified that the combination of Chr and BLG decreased the stability of BLG, while Bai had little effect on it. Moreover, the foaming properties of BLG got better in the presence of the two flavonoids compounds and Bai improved its emulsification stability of the protein, but Chr had the opposite effect. This work provides a new idea for the development of novel dietary supplements using functional proteins as flavonoid delivery vectors.

## 1. Introduction

Dietary supplements and functional foods are of growing interest to the public. From the perspective of food processing, the interactions between polyphenols and proteins play an important role in the stability of carrier systems [1,2]. Therefore, the behavior of the polyphenol–protein system has attracted more and more attention. Most of the molecular interactions have been reported to be non-covalent between phenolic compounds and proteins [3,4]. In the presence of enzymes or alkali or heat induction, phenolic compounds oxidized to quinones, where they covalently bound to proteins [2,5].

The non-covalent interactions between polyphenols and proteins mainly include hydrogen bonds, van der Waals forces, and hydrophobic and electrostatic interactions [6]. The formation of a hydrogen bond depends on the accessibility of the peptide bond and secondary structure of protein. Hydrophobic interactions are usually formed by interactions between benzene rings of polyphenol and hydrophobic side chains of amino acids [7].

β-lactoglobulin (BLG), the most abundant proteins in whey, has been extensively studied due to its outstanding physicochemical properties and excellent nutritional value [8]. Natural BLG (molecular weight 18.4 kDa) is a globular protein with 162 residues, and its hydrophobic core (calyx) is located in the middle of the β-barrel, which is composed of eight antiparallel β-strands and one α-helix [9]. Several studies have shown that there are two primary hydrophobic binding sites in the BLG molecule: one located inside the hydrophobic core, and another located on the surface between the β-barrel and α-helix [10]. BLG has been proved to have a strong affinity with vitamins, β-carotene, fatty acids, polyphenols, and other bioactive substances, and is an ideal ligand substance to transport bioactive substances and improve their bioavailability and stability [11,12].

Flavonoids, a kind of polyphenol natural active ingredient, are widely found in natural plants. The structure of flavonoids is constitutive of two benzene rings with phenolic hydroxyl groups connected to each other by a central three-carbon atom. The differences of physicochemical properties of flavonoids are mainly caused by the groups connected to them [12]. Baicalein (Bai) is a flavonoid extracted from the herb *Scutellaria baicalensis*
*Georgi* while chrysin (Chr) is abundant in honey and propolis (structure shown in Figure 1A,B), and both of them have important applications in traditional Chinese medicine. Previous studies have found that Bai and Chr exhibit a wide range of biological properties, including antioxidative, antiphlogistic, antimicrobial, antihypertensive, and anticancer activities [13,14,15]. However, their practical applications are greatly limited as a result of their poor aqueous solubility, poor absorption, rapid metabolism, and low bioavailability [16].

It was reported that non-covalent interactions between flavonoids and BLG are closely related to the structure of flavonoids. Such an interaction can alter the conformation of BLG. Li et al. reported that there were certain differences in the interactions between BLG and four structurally similar flavonoids (apigenin, naringenin, genistein, and kaempferol) [12]. Specifically, hydrogen bonds and van der Waals forces were the main driving forces in the binding of apigenin and naringenin to BLG, but hydrophobic interaction played a major role in the interaction of genistein or kaempferol with BLG. Jia et al. found that the binding affinity of epigallocatechin-3-gallate to BLG was stronger than that of chlorogenic acid and ferulic acid, and their binding altered the secondary structures of BLG as the transformation from α-helix to β-structures transition and led to the change in the surface hydrophobicity of BLG [17]. Yahyaei et al. reported a molecular dynamics (MD) simulation study of follicle stimulating hormone encapsulation in cholesterol-modified chitosan nanogels to investigate its potential molecular mechanisms [18]. As far as we know, the mechanism of Bai/Chr binding to BLG and influence on the functional properties of BLG are unclear, which limit the design of a FA delivery system using BLG as a carrier protein.

This work aimed to comprehensively study the potential binding mechanism between BLG and Bai/Chr and its impact on the structure and functional characteristics of BLG by multi-spectroscopic means along with molecular docking and MD simulation techniques. The results may provide a better understanding of how flavonoids behave when they interact with BLG and may be useful for the formation of a delivery system, beverage production, and exploitation in dietary supplements.

## 2. Materials and Methods

### 2.1. Materials

BLG (purity ≥ 90%) was purchased from Sigma–Aldrich Co. (St. Louis, MO, USA). Bai (purity ≥ 98%) was acquired from National Institutes for Food and Drug Control. Chr (purity ≥ 98%), 1-anilinonaph-thalene-8-sulfonic acid (ANS), and 5,5′-dithiobis-(2-nitrobenzoic acid) (DTNB) were obtained from Aladdin Chemical Co. (Shanghai, China). Soybean oil was purchased from Kerry Grains and Oils Co., Ltd. (Shenzhen, China). Other reagents used throughout the experiment were of analytical reagent grade.

### 2.2. Preparation of BLG and Flavonoid Solutions

The stock solution of BLG (20.0 µM) was dissolved in phosphate buffer (PBS, 50 mM, pH 6.8) and NaN_3_ (0.02%, *w*/*v*) was added as a preservative. The stock solutions of Bai and Chr (2.0 mM) were acquired by dissolving them in dimethyl sulfoxide (DMSO), respectively. The final DMSO concentrations were kept less than 2% to avoid the effect on the structure of BLG. All solutions were stored at 4 °C and refrigerated for later use.

### 2.3. Intrinsic Fluorescence Spectra

Intrinsic fluorescence spectroscopic test was measured on a F–7000 spectrophotometer (Hitachi, Japan) equipped with temperature controller according to the method described in the previous study [19]. Briefly, 3 mL solution of BLG (20.0 µM) was titrated by continuously adding different amounts of Bai or Chr (0–20.0 µM). The fluorescence spectra were recorded at different temperatures (298 K, 304 K, and 310 K) with the excitation wavelength of 280 nm and the excitation and emission slit set at 2.5 nm. Moreover, synchronous fluorescence spectra were conducted by setting the wavelength intervals (λ_em_–λ_ex_) at 15 nm and 60 nm, respectively. The 3D fluorescence spectra (molar ratios of BLG to Bai/Chr were 1:0 and 1:1) were recorded at excitation and emission wavelength range of 200−600 nm. All the fluorescence values were corrected to eliminate the internal filtering effect according to the previous method [20], and were used to evaluate the quenching mechanism of Bai/Chr binding to BLG by using the following equation [21]:(1)$$\frac{F_{0}}{F}=1+K_{q}\tau_{0}[Q]=1+K_{\mathrm{SV}}[Q]$$
where *F* and *F*_0_ are the fluorescence intensities of fluorophore with and without quencher; *K*_sv_ denotes the Stern–Volmer quenching constant; *K*_q_ is the quenching rate constant (*K*_q_ = *K*_sv_/*τ*_0_), and *τ*_0_ is the average lifetime of the biomolecule in the absence of a quencher (*τ*_0_ = 10^−8^  s); [*Q*] represents the experimental concentration of Bai/Chr.

For static fluorescence quenching mechanism, the binding constant (*K*_a_) and the number of binding sites (*n*) could be determined by the following equation [22]:(2)$$\log\frac{F_{0}-F}{F}=n\log K_{a}-n\log\frac{1}{[Q_{t}]-\frac{\left(F_{0}-F)[P_{t}\right]}{F_{0}}}$$
where [*Q*_t_] and [*P*_t_] are the concentrations of Bai/Chr and BLG, respectively.

### 2.4. Thermodynamic Parameters

In order to determine the primary forces between BLG and Bai/Chr, thermodynamic parameters were calculated using the following equations [23]:(3)$$\log K_{a}=-\frac{\Delta H{}^{\circ}}{2.303RT}+\frac{\Delta S{}^{\circ}}{2.303R}$$
(4)ΔG°=ΔH°−TΔS°
where *R* and *T* represent gas constant (8.314 J mol^−1^ K^−1^) and temperature (298, 304, and 310 K), respectively. Enthalpy change (Δ*H*°) and entropy change (Δ*S*°) of interaction between ligands and biomolecules are the main criteria for evaluating acting force, and free energy (Δ*G*°) is the main criteria for estimating the spontaneity of interaction.

### 2.5. Circular Dichromism (CD) Spectra

The CD spectra in the range of 200–250 nm were obtained from a CD spectrometer (MOS 450, Bio–Logic, Claix, France) equipped with 1.0 mm path length cuvette under the constant nitrogen flush. The solutions were prepared with molar ratios of Bai or Chr to BLG of 0:1, 1:1, and 2:1, respectively. The contents of secondary structure of BLG were determined based on our previously reported study [24].

### 2.6. Determination of BLG Free Sulfhydryl (SH) Content

The content of free SH was determined by spectrophotometry according to the report by Takić et al. with minor modifications [25]. Briefly, Bai/Chr concentrations were varied from 0 to 20.0 µM, and BLG was kept at 20.0 µM. Then the two solutions were incubated with 10.0 mM DTNB in 50 mM PBS (pH 6.8) for 30 min at room temperature. The absorbance of the sample at 412 nm was recorded and the molar extinction coefficient (14,360 M^−1^ cm^−1^) was obtained from the standard curve of L-cysteine, which was used to calculate the number of total sulfhydryl groups per gram of protein.

### 2.7. Measurement of Protein Surface Hydrophobicity (PSH)

PSH of BLG in the presence or absence of Bai or Chr was determined using ANS as a fluorescent probe. In the BLG solution (20.0 µM) alone or Bai/Chr–BLG mixture the ANS solution (5 mM) was continuously added until the fluorescence intensity of the mixture remained constant. The fluorescence intensity of the sample was recorded in the wavelength range of 400–600 nm upon 380 nm excitation. The PSH values were calculated by the following equations [26]:(5)$${\left[\mathrm{ANS}\right]}_{\mathrm{bound}}=F/B$$
(6)$${\left[\mathrm{ANS}\right]}_{\mathrm{free}}={\left[\mathrm{ANS}\right]}_{\mathrm{total}}-{\left[\mathrm{ANS}\right]}_{\mathrm{bound}}$$
(7)$$\frac{F}{{\left[\mathrm{ANS}\right]}_{\mathrm{free}}}=-\frac{F}{K_{d}^{\mathrm{app}}}-\frac{F_{\max}}{K_{d}^{\mathrm{app}}}$$
(8)$$\mathrm{PSH}=\frac{F_{\max}}{[\mathrm{BLG}]K_{d}^{\mathrm{app}}}$$
where [ANS]_free_ and [ANS]_bound_ are the concentration of free and bound ANS, respectively. $K_{d}^{\mathrm{app}}$ is the apparent dissociation constant of ANS, *F*_max_ is the maximum fluorescence intensity at saturating ANS concentration and *B* is the proportionality coefficient.

### 2.8. Computational Docking

Molecular docking method makes it more intuitive to predict the interaction between protein and ligand [27]. LibDock protocol in Discovery Studio (version 3.5), an application of a CHARMm-based docking tool using a rigid receptor protocol, was used to predict the binding posture of Bai or Chr to BLG [28]. The molecular docking was carried out according to the method of Song et al. [29]. The structure of BLG (PDB ID: 2Q2M) was acquired from the Protein Data Bank (http://www.rcsb.org/) (accessed on 10 August 2021) and the crystal structures of Bai and Chr were derived from the PubChem database. Briefly, pretreatment of BLG involved the removal of water molecules and the addition of hydrogen and polarity. CHARMm force field algorithm was added to calculate and order the energy of each pose, and the posture with the lowest interactive energy and the highest score was used as the optimal model for further analysis.

### 2.9. MD Simulation

MD simulation was conducted by using GROMACS package (version 5.1.2) and the AMBER99SB-ILDN force field was applied [30]. The BLG file used the same file from molecular docking and original water molecules were removed. Then, a dodecahedron solvent box was added where edges were defined 1.0 nm away from the protein surface in 3 coordinate directions. Next, 36,158 water molecules were gained by using the simple point charge (SPC) model, and 6 Na^+^ were added to neutralize the charge of the system. The topological parameters of BLG were created by GROMACS package, and the topological parameters of Bai and Chr were obtained from the Acpype Sever (http://bio2byte.be/acpype/) (accessed on 23 July 2021). Periodic boundary conditions were utilized throughout the simulation. The steepest descent algorithm was adopted to minimize the energy by 10,000 steps [31]. During the NVT and NPT ensembles, the temperature and pressure were equilibrated to 300 K and 1 bar, respectively [32]. Finally, a time step of 2 fs was used. The free BLG and flavonoids–BLG systems were simulated for 25 ns by the leapfrog approach [33].

### 2.10. Determination of Foaming Properties

Foaming ability (FA) and foaming stability (FS) were measured as described by Wang et al. with some modifications [34]. A certain amount of BLG solution (1.0 mg mL^−1^) was added to a 100 mL cylinder, and Bai or Chr was added to form solutions with different ratios of BLG to Bai or Chr (40:1, 30:1, 20:1, 10:1). The initial height *H*_0_ of the foam was recorded. The foam was then sheared at 10,000 rpm in a high-speed dispersion machine for 1 min and the height of the foam was recorded as *H*_1_. The height of the foam was recorded as *H*_2_ 30 min later. The FA (%) and FS (%) were calculated according to the following equations:(9)$$\mathrm{FA}\;(\%)=\frac{H_{1}-H_{0}}{H_{0}}\times 100$$
(10)$$\mathrm{FS}\;(\%)=\frac{H_{2}-H_{0}}{H_{1}-H_{0}}\times 100$$

### 2.11. Determination of Emulsifying Properties

Emulsifying ability (EA) and emulsifying stability (ES) were investigated by the method of Xiong et al. with slight modification [35]. BLG solution (1 mg mL^−1^) was added to Bai/Chr to form the solutions with different ratios of BLG to Bai/Chr (40:1, 30:1, 20:1, 10:1), and then soybean oil was added to form BLG solution containing 10% (*v*/*v*) oil. The emulsion was fully dispersed for 2 min at a rate of 7000 r min^−1^ and repeated for three times. At 0 and 10 min, 30 µL of emulsion was extracted from the bottom of the sample and immediately added to 3 mL of 0.1% SDS solution. The absorbance of the sample was measured at 500 nm and recorded as *A*_0_ and *A*_10_, respectively. The EA (m^2^ g^−1^) and ES (min) were calculated according to the following equations:(11)$$\mathrm{EA}\;(m^{2}{\;g}^{-1})=\frac{2\times 2.303}{c\times \Phi \times L\times 10^{4}}\times A_{0}\times D$$
(12)$$\mathrm{ES}\;(\min)=\frac{A_{0}}{A_{0}-A_{10}}\times 10$$
where *c* is the weight of the BLG per unit volume (g mL^−1^), Φ is the volume fraction of the oil phase, *L* is the length of light path, and *D* is the dilution factor, *D* = 100.

### 2.12. Statistical Analysis

All measurements were carried out in triplicate and the values were described as mean ± standard deviation. Data were conducted by one-way ANOVA, followed by Duncan’s test at a confidence level of 0.05.

## 3. Results and Discussion

### 3.1. Fluorescence Quenching

The intrinsic fluorescence of BLG was mainly contributed by two tryptophan (Trp) and four tyrosine (Tyr) residues, which were activated and quenched at 340 nm after excitation at 280 nm [36]. This feature has been used by many researchers to determine the binding strength of flavonoids and BLG [37]. As shown in Figure 1A,B, the fluorescence intensity of BLG gradually weakened with the titration of Bai or Chr. However, the quenching abilities of Bai and Chr were different. After 10 times titration, Bai and Chr led to 63.2% and 59.0% BLG fluorescence quenching at 298 K, respectively. These results indicated that both Bai and Chr interacted with BLG, but the interaction of Bai was stronger than that of Chr, which may be because Bai has one more hydroxyl group than Chr [38].

The fluorescence quenching types are classified as dynamic quenching and static quenching [39]. For dynamic quenching, increasing temperature may accelerate the collision between the quencher and fluorophore due to higher diffusion velocity, thus the quenching constant may increase. While for static quenching, ligand–protein complex may become more unstable and the quenching constant may decrease when temperature increases [40]. As shown in Figure 1C,D, the Stern–Volmer plots displayed good linearity at the three different temperatures, indicating that Bai or Chr quenched the fluorescence of BLG by one single quenching mechanism [41]. As listed in Table 1, their *K*_sv_ values decreased with rising temperature (*p* < 0.05), indicating that the quenching mechanism was static. Moreover, the calculated *K*_q_ values were two orders of magnitude greater than the maximum diffusion collision quenching constant value (2.0 × 10^10^ L mol^−1^ s^−1^), further proving that the quenching process of BLG by Bai or Chr was static quenching.

In the case of static quenching, the values of *K*_a_ and *n* between Bai or Chr and BLG were computed based on the double logarithmic equation (Equation (2)). The *n* values for the binding of Bai or Chr with BLG were approximately 1 (Table 1), indicating that there was only one binding site in BLG for Chr or Bai. The *K*_a_ values of Bai or Chr binding to BLG at 298 K were 2.78 × 10^5^ and 2.14 × 10^5^ L mol^−1^, respectively. This outcome indicated that there was strong binding affinity in the complexes, and the interaction between BLG and Bai was stronger than that of Chr. Xiao et al. pointed out that the effectiveness of biologically active ingredients largely depends on their level of binding to the protein being delivered [42]. Aditya et al. reported that curcumin nanoemulsion improved the bioavailability of curcumin using BLG as a delivery carrier, and the binding affinity of curcumin with BLG was 4.04 × 10^−5^ L mol^−1^ [43,44]. Some similar studies on the combination of BLG and tea polyphenols have also been reported including EGCG, GCG, etc. [45,46]. Therefore, as an excellent carrier protein, BLG may have the potential to improve the bioavailability of Bai/Chr.

### 3.2. Thermodynamic Analysis

The thermodynamic parameters including Δ*H*°, Δ*S*° and Δ*G*° were determined by Van’t Hoff equation Equations (3) and (4) and used to evaluate the dominant binding forces. As shown in Table 1, the Δ*G*° values were negative for the interaction of Bai or Chr with BLG, which meant that the binding process of Bai or Chr to BLG was spontaneous. The values of Δ*H*° and Δ*S*° were −5.59 kJ mol^−1^ and 85.44 J mol^−1^ K^−1^ for Bai–BLG interaction and 6.21 kJ mol^−1^ and 122.18 J mol^−1^ K^−1^ for Chr–BLG binding, respectively, indicating that hydrogen bonds and hydrophobic interactions were the dominating forces in the Bai–BLG binding, while hydrophobic forces played important roles in the Chr–BLG complex [47].

### 3.3. Synchronous Fluorescence Spectra

Synchronous fluorescence spectroscopy can reflect the microenvironmental, and the polarity change around the Tyr and Trp residues by setting interval (λ_em_–λ_ex_) were 15 and 60 nm, respectively. With gradual addition of Bai or Chr, the fluorescence intensity of BLG reduced and the maximum emission wavelength of Trp (Figure 1G,H) residues showed significant red-shift but no change for that of Tyr (Figure 1E,F) residues. These results indicated that the polarity of Trp residues microenvironment was enhanced by both Bai and Chr. We inferred that both Bai and Chr could change the conformation of BLG, making the microenvironment near Trp residues more hydrophilic, while Tyr residues remained almost unchanged. Similar behaviors have been found in the interactions of flavonoids like apigenin, naringenin, genistein, and kaempferol with BLG and α-lactalbumin [12,37]. These studies found that the flavonoids could alter the conformation of α-lactalbumin or BLG, and the Trp residues have the tendency to be exposed to more hydrophilic environments. In addition, no obvious difference for the ratio of synchronous fluorescence quenching (RSFQ = 1 − *F*/*F*_0_) of Tyr and Trp by Bai or Chr was observed, indicating that Tyr and Trp contributed almost equally to the fluorescence quenching.

### 3.4. Three-Dimensional Fluorescence Spectra

Three-dimensional (3D) fluorescence spectrometry is an effective method to provide thorough information of conformational change of proteins [20]. The conformational change of BLG induced by Bai or Chr is displayed in Figure 2. Peak a is the Rayleigh scattering peak of BLG (λ_ex_ = λ_em_). Peak 1 represents the fluorescence behavior of tyrosine and tryptophan residues, and Peak 2 mainly characterizes the fluorescence characteristics of the main chain structure of BLG polypeptide [48]. It can be seen that the fluorescence intensities of Peak 1 and 2 went down after addition of Bai (Figure 2C,D, from 522.1 to 395.4, and from 430.6 to 206.8, respectively) or Chr (Figure 2E,F, from 522.1 to 378.5, and from 430.6 to 205.3, respectively). The attenuation of Peak 1 and Peak 2 indicated that there was molecular interaction between the two flavonoids and BLG, which may alter the conformation of BLG [18]. Moreover, the locations of Peak 1 and 2 changed slightly, indicating that the interactions might induce partial unfolding of the protein polypeptides. These results were consistent with a report that the interaction of some plant polyphenol compounds with globular proteins caused the structural unfolding of the proteins [49].

### 3.5. CD Spectra

CD spectroscopy is an effective method to monitor protein structural changes, so the conformational changes of BLG in the presence of Bai or Chr was detected by CD spectra. The typical characteristic of the β-sheet structure of BLG was a single negative peak at 218 nm on CD spectrum [50]. As shown in Figure 3A,B, with the increase of Bai or Chr concentration, the negative ellipticity at 218 nm tended to decrease (curves 1 → 3). The content of secondary structure of BLG was summarized in Table 2. The contents of α-helix and β-sheet tended to decrease in both Bai–BLG and Chr–BLG systems. Specifically, when the molar ratios of Bai/Chr to BLG increased from 0:1 to 2:1, the attenuation of α-helix content in Chr–BLG system (from 19.21% to 11.25%) was greater than that in Bai–BLG system (from 19.21% to 15.85%) (*p* < 0.05). Similar results were observed in β-sheet content of Chr–BLG system (from 39.46% to 33.06%) and Bai–BLG system (from 39.46% to 34.12%) (*p* < 0.05). These changes in the secondary structure indicated that the interaction between Bai/Chr and BLG resulted in the partial unfolding of BLG structure, which caused the structure of the protein to become looser. Chr had a more obvious impact on the structure of BLG than Bai, which may also lead to differences in the functional properties of BLG.

### 3.6. Free SH Group Change in BLG

SH group is one of the main groups of protein, which has an important influence on the functional properties of protein [51]. At the same time, free SH group has a certain antioxidant activity which may protect proteins from oxidative damage and participate in maintaining the balance of redox reaction in vivo [52]. Generally, the free SH group is near the hydrophobic area of protein, and the SH content of protein is closely related to the hydrophobicity of protein surface [53]. As shown in Figure 3C, the contents of free SH group of BLG gradually weakened with the enhancement of Bai/Chr concentration, suggesting that the interaction between Bai/Chr and BLG may cause the shrinking of BLG peptide chain and hiding of the hydrophobic area of BLG, which led to a decrease in SH content of BLG.

### 3.7. PSH Change of BLG

PSH changes the structure and interface properties of proteins and further affects the functional properties of proteins, which is one of the basic properties of food components [54]. It was reported that the endogenous fluorescence intensity of ANS was weak, but when ANS bound to hydrophobic amino acids on protein surface the fluorescence intensity was significantly increased [26]. The fluorescence intensity of the two systems showed a typical hyperbolic curve (Figure 4A,B), suggesting that ANS acted on the surface hydrophobic group of BLG. With the increase of ANS concentration, the fluorescence intensity of BLG increased and reached saturation. In the presence of different concentrations of Bai/Chr, the increase of fluorescence intensity of the two systems was significantly (*p* < 0.05) lower than that of BLG. The values of $K_{d}^{\mathrm{app}}$ and *F*_max_ were obtained from the Scatcherd plot (Figure 4C,D), and the results are summarized in Table 3. These results indicated that both Bai and Chr could reduce the PSH value of BLG (*p* < 0.05). This may be because the binding of Bai/Chr with BLG resulted in shrinking of the polypeptide chain of the secondary structure and hiding of the hydrophobic regions of BLG.

### 3.8. Molecular Docking

The LiDock algorithm of Discovery Studio was used to simulate the molecular docking between Bai or Chr and BLG. There are three binding sites in BLG for ligand binding. Site I and Site II were considered as the most favorable binding sites [50,55]. Site I and site II were situated in the hydrophobic cavity and on the surface of BLG, respectively. In the study, the most favorable results with the highest LibDock score were in Site II (LibDock scores were 89.80 and 91.67 for Bai–BLG system and Chr–BLG system, respectively). Therefore, Site II was used as the most possible binding site. It was observed that both Bai and Chr interacted with the residues on the surface of BLG instead of the cavity **(**Figure 5A,B). The residues including Trp19, Tyr20, Ser21, Leu22, Tyr42, Val43, Glu44, Gln59, Val123, Thr125, Pro126, Glu127, Phe151, and Asn152 participated in the formation of Van der Waals forces, hydrophobic interactions, and hydrogen bonds in Bai-BLG complex (Figure 5C,E). While the residues including Trp19, Tyr20, Ser21, Leu22, Tyr42, Val43, Glu44, Gln59, Val123, Thr125, Pro126, Phe151, and Asn152 took part in the formation of van der Waals forces and hydrophobic interactions in Chr–BLG complex (Figure 5D,F). In particular, two hydrogen bonds (Gln59 and Tyr42) were generated in the Bai–BLG complex, while no hydrogen bond was formed in the Chr–BLG interaction. Therefore, there was stronger binding affinity in the Bai–BLG interaction than that of Chr–BLG binding. Free energy decomposition results were performed using molecular mechanics/Poisson–Boltzmann surface area (MM–PBSA) to analyze the contribution of amino acid residues in the interactions between two flavonoids and BLG [29]. As shown in Table 4, compared with the Chr–BLG complex (−12.89 ± 0.26 kJ mol^−1^), the total energy of the Bai–BLG complex (−46.56 ± 0.93 kJ mol^−1^) was significantly lower (*p* < 0.05), indicating that the binding of Bai with BLG was stronger than that of Chr and BLG. These findings supported the above fluorescence quenching results that the binding affinity of Bai to BLG was stronger than that of Chr.

### 3.9. MD Simulation

Molecular docking results obtained by LibDock protocol are mainly about the information on the interaction between rigid protein and flavonoids. Therefore, MD simulation is needed to obtain the complementary information of the binding between flexible proteins with flavonoids. In order to simulate the dynamic properties of Bai–BLG and Chr–BLG complexes and verify the results obtained from molecular docking, MD simulation was performed by GROMACS (version 5.1.2) in the Linux operating system.

Root mean square deviation (RMSD) values were used to determine the time for the system to reach a stable state and evaluate the stability of the two complexes compared with free BLG [56]. As shown in Figure 6A,B, the RMSD values of free BLG displayed steady state after 3 ns. Both Bai–BLG and Chr–BLG systems reached a relative stable state around 12 ns. In Chr–BLG system, a great fluctuation appeared in the time range of 4–11 ns, and the maximum atomic undulation of RMSD reached 0.34 nm, but the Bai–BLG system exhibited a relatively small disturbance in this time range, with the maximum RMSD value of 0.25 nm. We inferred that the progress of Chr entering the hydrophobic area of BLG was more violent than Bai. Hence, the final RMSD values of Chr–BLG system were much higher than those of Bai–BLG system, and the Chr–BLG complex was considerably less stable. Al-Shabib et al. concluded that the RMSD of protein and ligand fluctuated greatly in the initial period, which may be caused by the ligand entering the hydrophobic cavity of protein [50]. Interestingly, after Bai and Chr bound to BLG, the fluctuation of complexes increased, and the stability of protein reduced. Root mean square fluctuation (RMSF) was used to simulate the root mean square shift of the system and to investigate the amino acid residue flexibility of BLG [4]. As illustrated in Figure 6C, the RMSF values of Bai–BLG, Chr–BLG, and free BLG were in the order of Chr–BLG > Bai–BLG > BLG. The combination of both ligands induced a high degree of flexibility in protein structure. Specifically, in the Bai–BLG system, compared with free BLG, the residues 40–60, 125–150 fluctuated more, while in the Chr–BLG system, the residues 60–75, 82–95, and 105–118 showed greater fluctuations. Obviously, the fluctuation extent of the Chr–BLG system was larger than that of Bai–BLG. This may be due to the slight difference in the structure of the ligand and the way it interacted with BLG. The binding of Bai/Chr to BLG had no significant effect on the solvent accessible surface area (SASA) of BLG (Figure 6D,G). Therefore, we focused on specific fluorophores (Trp and Tyr residues), and they were selected to make the SASA diagram. As shown in Figure 6E,F,H,I, there was no significant difference in the SASA of Tyr residues in both complexes, which was in line with the results of synchronous fluorescence, suggesting that the microenvironment of Tyr20, Tyr42, Tyr99, Tyr102 residues did not significantly change. In the case of Trp residues of both complexes, there was an obvious increase in the SASA of Trp19 and Trp61 residues compared to those of free BLG, meaning that the microenvironment of these two Trp residues became more hydrophilic. This result supported that of synchronous fluorescence [57].

### 3.10. Foaming Property Change of BLG

As shown in Figure 7A,B, compared with the FA of free BLG (25.2%), FA increased by 18.08% and 51.32%, respectively, as the molar ratios of BLG to Bai/Chr changed from 40:1 to 10:1 (*p* < 0.05). Although there was no concentration dependence, the FA values of BLG in the system with flavonoids were larger than that of the control group (BLG). Meanwhile, the addition of Bai/Chr significantly enhanced the FS of BLG. Compared with the BLG system alone, the FS of the Bai/Chr–BLG system (the molar ratio of BLG to Bai/Chr was 10:1) increased by 80.64% and 85.57%, respectively (*p* < 0.05). These outcomes indicated that flavonoid treatment improved the foaming performance of BLG. This could be due to the expansion of the molecular structure of BLG increasing the flexibility of the molecule and promoting the improvement of the FA of BLG [5]. It also may be that flavonoids promoted protein cross-linking to cause protein to be more adsorbed on the water–air interface, thereby improving the FS of BLG [58].

### 3.11. Emulsifying Property Change of BLG

EA and ES are important indicators of protein functional properties. EA reflects the speed and ability of protein unfolding at the oil–water interface, while ES reflects its ability to form a rigid structure to protect oil droplets [35]. As shown in Figure 7C,D, with the increase of Bai or Chr concentration, the EA of BLG slightly improved (*p* < 0.05). Interestingly, Bai promoted the enhancement of ES of BLG, while Chr did the opposite (*p* < 0.05). For the Chr–BLG system, more hydrophobic amino acids were exposed on the surface due to the expanded structure, resulting in greater EA than that of Bai–BLG system [59]. The presence of Chr promoted the increase of the hydrophobicity of BLG, which induced BLG to aggregate more easily and the surface tension of the emulsion to increase, which eventually reduced the ES of BLG [60]. Moreover, the addition of Chr induced the structure of BLG to be more unstable. The effect stacked up and eventually led to the decrease of ES in the Chr–BLG system. However, the result of Bai was opposite. Compared with Chr, there was no great fluctuation in the binding process of Bai and BLG, which was the possible reason for the ES increase of the Bai–BLG system.

## 4. Conclusions

This work studied the interaction of Bai and Chr with BLG and their effects on the functional properties of BLG. The fluorescence quenching mechanism of BLG by Bai or Chr was static. Hydrophobic interaction and hydrogen bond play an important role in the formation of the Bai/Chr–BLG complex. Compared with Chr, Bai had one more hydroxyl group to form more hydrogen bonds with BLG, thus the affinity for Bai binding to BLG was stronger. The protein surface hydrophobicity and free sulfhydryl group of BLG were decreased in the presence of the two flavonoids. Both Bai and Chr caused the attenuation in the content of α-helix, resulting in a loose structure of BLG. Molecular docking proved that the binding site was located on the surface of BLG. The residues Tyr20 and Phe151 contributed the most to the binding of Bai with BLG, while Asn152 contributed the most in Chr–BLG interaction. RMSD values of MD simulation demonstrated that the combination of Chr and BLG reduced the stability of BLG, while Bai had a minor effect. The analysis of synchronous fluorescence and SASA indicated that the polarity of the Trp microenvironment in the complexes was enhanced. The two flavonoids had positive effects on the foaming properties of BLG, and Bai enhanced the emulsifying stability of BLG while Chr impaired it. These findings may contribute to the understanding of the interaction mechanism of Bai/Chr and BLG, expand the application range of functional proteins, and provide a theoretical basis for the development of novel dietary supplements using functional proteins as flavonoid carriers.

## Figures and Tables

**Figure 1 foods-11-00165-f001:**
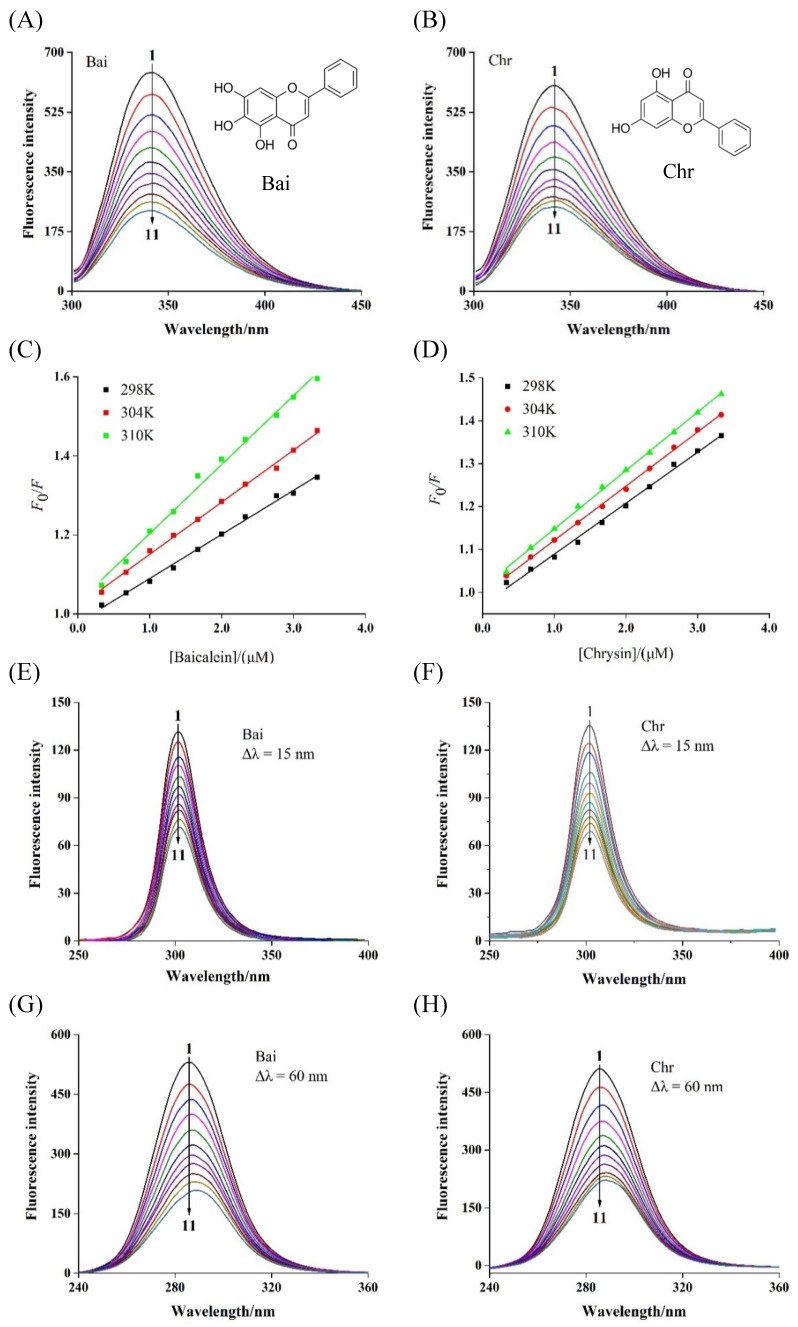
Fluorescence spectra of BLG in the presence of Bai (**A**) and Chr (**B**) at various concentrations, and the Stern–Volmer plots (**C**,**D**) at different temperatures (*T* = 298 K, 304 and 310 K, pH 6.8, λ_ex_ = 280 nm). *c*(BLG) = 20 µM, *c*(Bai/Chr) = 0, 2.0, 4.0, 6.0, 8.0, 10.0, 12.0, 14.0, 16.0, 18.0, 20.0 µM for Curves 1 → 11, respectively. Synchronous fluorescence spectra of BLG in the absence and presence of Bai and Chr (pH 6.8, *T* = 298 K). Δ*λ* = 15.0 nm (**E**,**F**), Δ*λ* = 60.0 nm (**G**,**H**). *c*(BLG) = 20.0 µM, *c*(Bai/Chr) = 0 to 20.0 μM for Curves 1 → 11, respectively.

**Figure 2 foods-11-00165-f002:**
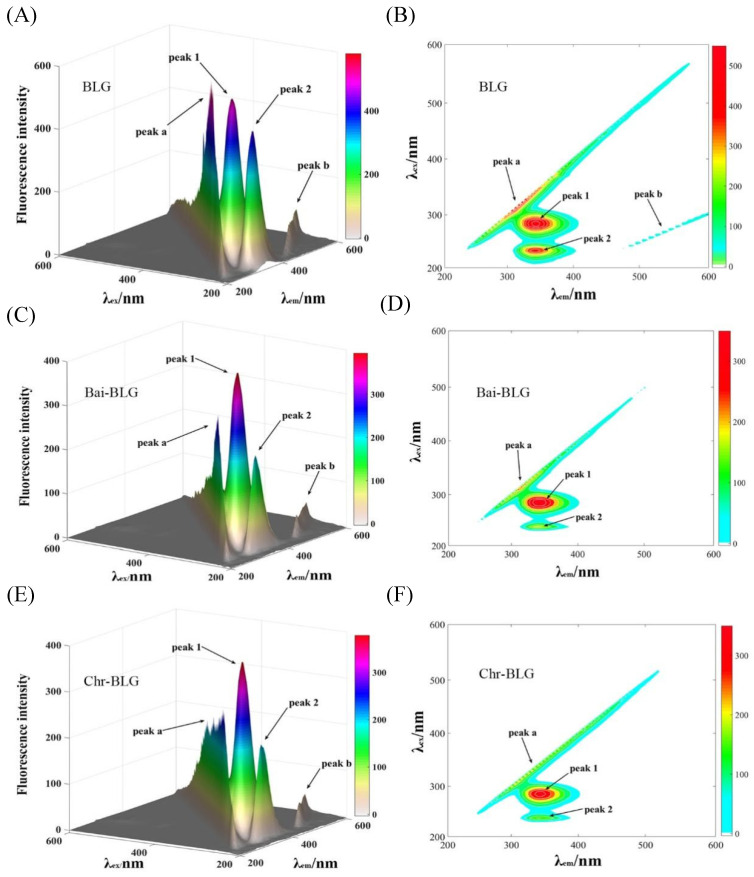
3D fluorescence spectra of free BLG (**A**), Bai–BLG (**B**), and Chr–BLG (**C**) systems. Contour maps of free BLG (**D**), Bai–BLG (**E**), and Chr–BLG (**F**) systems. *c*(BLG) = 20.0 µM, *c*(Bai/Chr) = 20.0 μM.

**Figure 3 foods-11-00165-f003:**
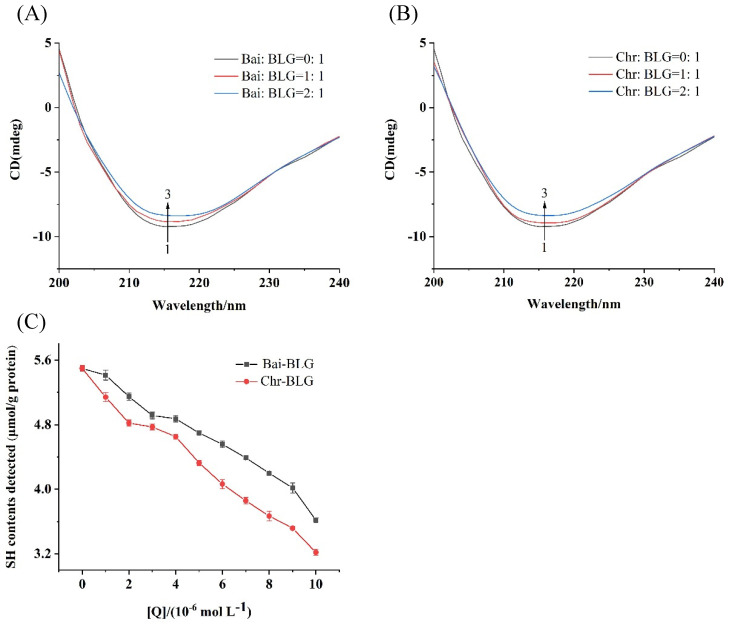
Far–UV CD spectra of BLG with increasing concentrations of Bai (**A**) and Chr (**B**), the molar ratios of Bai/Chr to BLG were 0:1, 1:1 and 2:1. (**C**) Effect of different concentrations of Bai/Chr on sulfhydryl group (SH) contents of BLG.

**Figure 4 foods-11-00165-f004:**
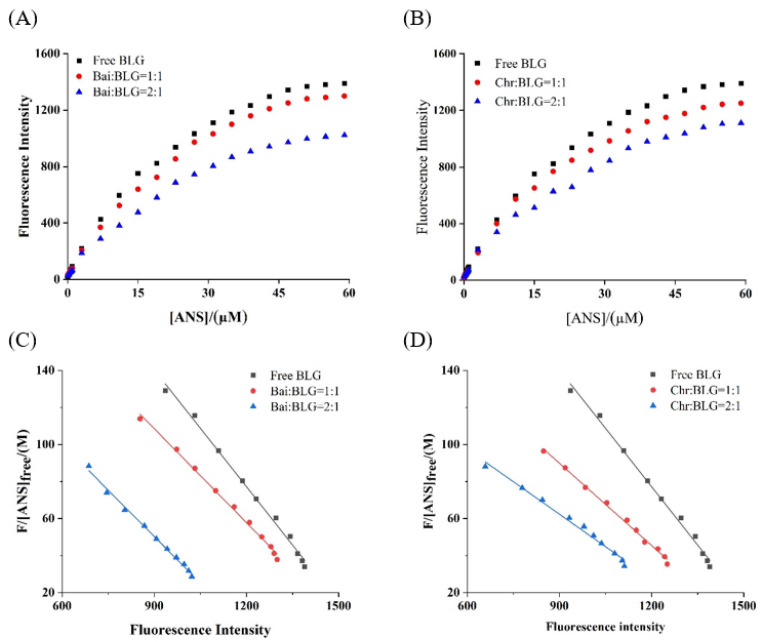
Binding of ANS to BLG in the absence and presence of Bai (**A**) and Chr (**B**). *c*(BLG) = 20.0 µM. The molar ratios of BLG to Bai or Chr were 1:0, 1:1 and 1:2. Scatcherd plots for the titration with increasing concentrations of ANS relative to BLG in the absence and presence of Bai (**C**) and Chr (**D**).

**Figure 5 foods-11-00165-f005:**
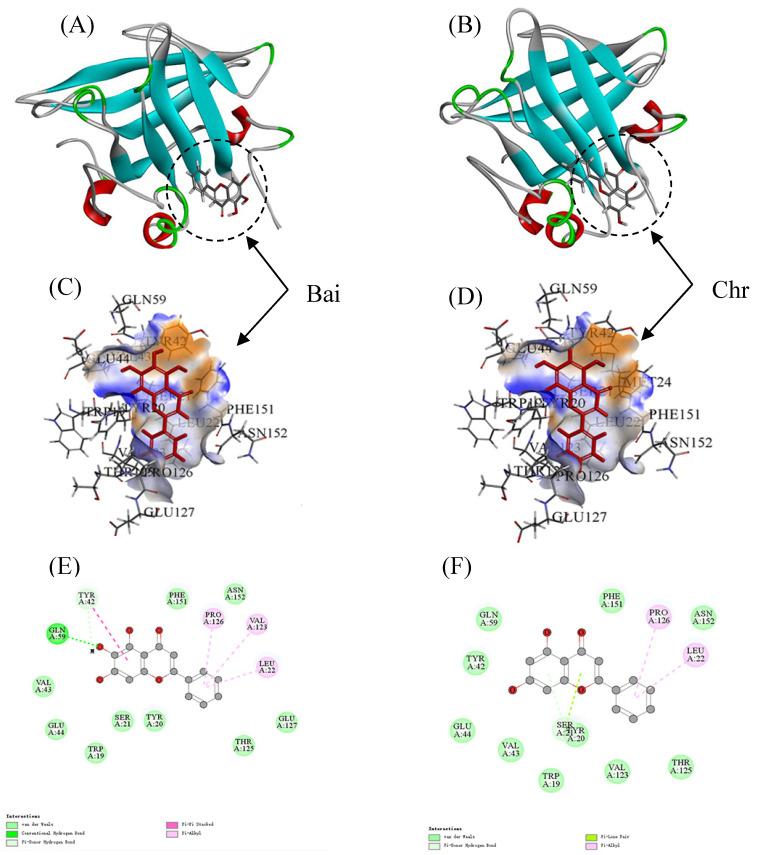
Docking results between flavonoid (Bai or Chr) and BLG by LibDock of Discovery Studio. 2D and 3D diagrams of Bai (**A**,**C**,**E**) and Chr (**B**,**D**,**F**) interacting with BLG.

**Figure 6 foods-11-00165-f006:**
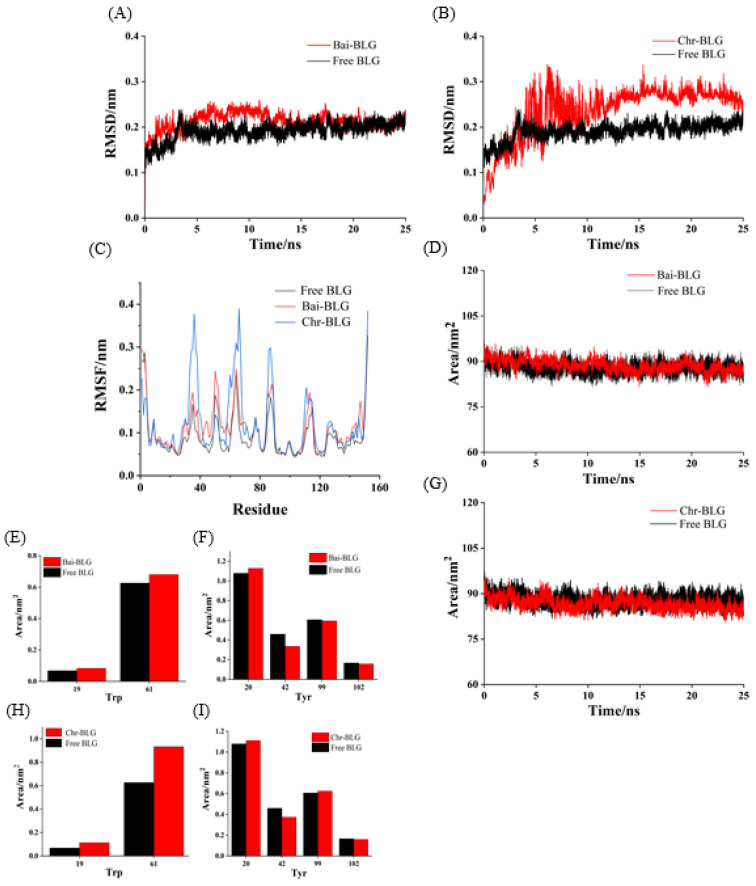
MD simulation of Bai/Chr with BLG for 25 ns. The RMSD (**A**,**B**) and RMSF (**C**) plots and SASA (**D**,**G**) of the complex and free BLG. The SASA of residues Trp (**E**,**H**) and Tyr (**F**,**I**).

**Figure 7 foods-11-00165-f007:**
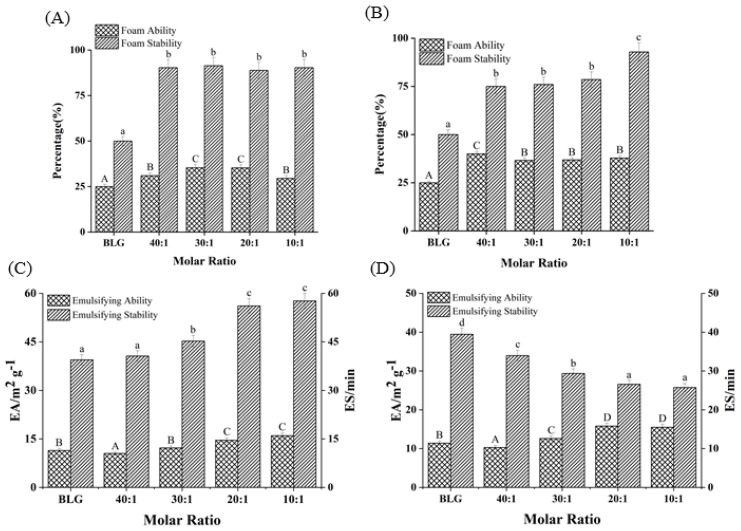
Effects of different molar ratios of Bai–BLG (**A**) and Chr–BLG (**B**) systems on foam ability and foam stability. Effects of different molar ratios of Bai–BLG (**C**) and Chr–BLG (**D**) systems on the emulsifying ability and emulsifying stability. The molar ratios of BLG to Bai or Chr were 40:1, 30:1, 20:1, and 10:1.

**Table 1 foods-11-00165-t001:** Quenching constant (*K*_SV_), binding constant (*K*_a_), the number of binding site (*n*), and the thermodynamic parameters of Bai–BLG and Chr–BLG complexes at three different temperatures.

System	*T* (K)	*K*_sv_ (×10^4^ L mol^−1^)	*R* ^a^	*K*_a_ (×10^5^ L mol^−1^)	*n*	*R* ^b^	Δ*H*° (kJ mol^−1^)	Δ*G*° (kJ mol^−1^)	Δ*S*° (J mol^−1^ K^−1^)
Bai–BLG	298	5.75 ± 0.01 ^a^	0.9955	2.78	1.03 ± 0.03	0.9971	−5.59	−31.05	85.44
	304	4.76 ± 0.03 ^b^	0.9989	2.67	1.23 ± 0.02	0.9974		−31.56	
	310	4.47 ± 0.06 ^c^	0.9973	2.55	1.31 ± 0.03	0.9964		−32.07	
Chr–BLG	298	3.10 ± 0.04 ^d^	0.9986	2.14	1.45 ± 0.01	0.9921	6.21	−30.41	122.18
	304	2.94 ± 0.06 ^e^	0.9972	2.08	1.01 ± 0.02	0.9976		−31.14	
	310	2.59 ± 0.04 ^f^	0.9974	1.95	1.29 ± 0.01	0.9984		−31.88	

^a^*R* is the correlation coefficient for the *K_SV_* values, ^b^
*R* is the correlation coefficient for the *K_a_* values. Different letters indicate significant difference (*p* < 0.05).

**Table 2 foods-11-00165-t002:** Secondary structure contents of Bai–BLG and Chr–BLG systems at different molar ratios of Bai/Chr to BLG.

System	Molar Ratio	α-Helix/%	β-Sheet/%	β-Turn/%	Random Coil/%
Bai–BLG	0:1	19.21 ± 0.43 ^a^	39.46 ± 0.32 ^a^	22.61 ± 0.34 ^b^	18.72 ± 0.62 ^d^
	1:1	14.12 ± 0.70 ^c^	36.25 ± 0.25 ^b^	23.16 ± 0.12 ^a^	26.47 ± 0.41 ^b^
	2:1	15.85 ± 0.93 ^b^	34.12 ± 0.34 ^d^	24.77 ± 0.54 ^b^	25.26 ± 033 ^c^
Chr–BLG	0:1	19.21 ± 0.42 ^a^	39.46 ± 0.32 ^a^	22.61 ± 0.34 ^b^	18.72 ± 0.62 ^d^
	1:1	14.30 ± 0.47 ^c^	35.18 ± 0.12 ^c^	24.56 ± 0.48 ^a^	25.96 ± 0.34 ^bc^
	2:1	11.25 ± 0.58 ^d^	33.06 ± 0.30 ^e^	24.53 ± 0.93 ^a^	31.16 ± 0.10 ^a^

Different letters in the same column indicate significant difference (*p* < 0.05).

**Table 3 foods-11-00165-t003:** Effect of Bai/Chr on surface hydrophobicity of BLG at different molar ratios of BLG to Bai/Chr.

System	Molar Ratio	$K_{d}^{\mathrm{app}}\;(\times 10^{-6}\;L\;{\mathrm{mol}}^{-1})$	*F* _max_	PSH	Reduction Rate (%)
Bai–BLG	0:1	4.76 ± 0.01 ^d^	156.80 ± 3.14 ^a^	329.41 ± 6.59 ^a^	
	1:1	5.92 ± 0.12 ^ab^	154.31 ± 3.09 ^ab^	260.66 ± 5.21 ^b^	20.87 ± 0.42 ^d^
	2:1	6.00 ± 0.47 ^a^	120.27 ± 2.41 ^c^	200.45 ± 4.01 ^d^	39.15 ± 0.78 ^c^
Chr–BLG	0:1	4.76 ± 0.01 ^d^	156.80 ± 3.14 ^a^	329.41 ± 6.59 ^a^	
	1:1	5.67 ± 0.31 ^c^	151.76 ± 3.04 ^b^	252.82 ± 5.07 ^c^	23.25 ± 0.47 ^b^
	2:1	5.70 ± 0.10 ^bc^	110.17 ± 2.20 ^d^	193.69 ± 3.87 ^e^	41.20 ± 0.82 ^a^

Different letters in the same column indicate significant difference (*p* < 0.05).

**Table 4 foods-11-00165-t004:** Summary of free energy decomposition (kJ mol^−1^).

Energy Components	Bai–BLG Binding	Chr–BLG Binding
Van der Waals	−44.37 ± 0.89	−13.40 ± 0.27
Electrostatic	−17.84 ± 0.18	−3.37 ± 0.03
Polar	20.69 ± 1.03	5.72 ± 0.29
Nonpolar	−5.04 ± 0.10	−1.84 ± 0.04
Total	−46.56 ± 0.93 ^b^	−12.89 ± 0.26 ^a^

Different letters in the same column indicate significant difference (*p* < 0.05).

## Data Availability

Not applicable.

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
