# Peer review of "Effects of Baicalein and Chrysin on the Structure and Functional Properties of β-Lactoglobulin"

_foods, 2022, doi:10.3390/foods11020165_

Round 1

Reviewer 1 Report

  1. Why authors selected baicalein and chrysin with very similar structure for this study? They could use the glucuronide of baicalein, baicalin, instead of baicalein and assay its effects.
  2. In introduction, line 55, authors mentioned that “Baicalein (Bai) is a flavonoid extracted from the herb Scutellraria violacea Georgi” however baicalein extracted from Scutellaria genus mainly Scutellaria baicalensis Georgi, not only Scutellraria violacea, correct it.
  3. The authors used RMSD values to determine the stability of the chrysin and baicalein-combined BLG compared with free BLG, however there are different experimental methods such as equilibrium denaturation experiments to quantify how the presence of other compounds affect protein stability. These experiments yield insights into the forces driving protein folding and misfolding dynamics. It is suggested that authors apply an experimental procedure to evaluate the stability of BLG in the presence of chrysin and baicalein.
  4. It is suggested that authors evaluate the linkage of chrysin induced instability of BLG with its functions.
  5. The references cited are in many instances old (references 9, 11, 16, 32, 33, 35, 46 and 52), it is of utmost importance to cite newest possible references, and not the ones that are 20-25 or 30 years old. Majority of cited publications must be published in the last 10 years meaning from 2011 to 2021. Exchange the old references with new ones.
  6. Most of the provided figures in the paper do not have standard resolution, try to improve the quality of your figures with an online program, such as pixlr.com
  7. In line 256-257 authors assumed that the combination of baicalein with BLG could provide a stronger delivery capacity, improved absorption and bioavailability in vivo for Bai, however they did not perform a relevant assay to determine its bioavailability after contribution, provide a logical assay for confirmation.
  8. Italicize “in vitro” throughout the manuscript including References.

Reviewer 2 Report

The manuscript contents are relevant and of high interest for the community working on a supplement delivery system. Indeed the works explore the binding mechanism between BLG and two flavonoids, namely baicalein (Bai) and chrysin (Chr). In addition, the experimental and molecular modelling studies are reported and compared, making the paper attractive.

However, some critical shortcomings were detected and forced me to ask for a significant revision which I believe the Authors should be able to fix:

  1. In the materials section, all used reagents should be listed.
  2. Methods should be carefully reported,  e.g. line 101 "adding different amount of BAi and Chr"... specify the amount of BAi and Chr used and the buffer you use to make the solutions. You may wish to add supplementary materials to report the exact protocols you used.
  3. Table 1: what n stands for?
  4. Line 255: add sentences to explain the delivery mechanism/ capacity you suggested. Please, justify the increase in bioavailability following absorption of Bai-BLG or Chr-BLG complex. 
  5. CD spectra analysis: this paragraph needs to explain and comment more on the reported results. Also, please carefully check the figures cross-reference and captions. Finally, rescale the figure to visualise better the results you are reporting.
  6. Line 329-330: briefly explain which is/are the important effect(s) on the functional properties of the protein.
  7. Line 334-337: What else might induce a weakened fluorescence/signal? Were any control used in the experiments? Please, specify this in the methods?
  8. Table 3: is the molar ratio correctly expressed and reported? I was expected to find a Bai-BLG (0:1) and Bai-BLG (1:0). Same for Chr.
  9. I suggest the authors merge/overlay figures 6B and D for better visualisation. Or keep them separated, but highlight the residues' fluctuations using different colours ( and adjust the legend accordingly).
  10. Lines 462-464: As the Authors wrote, this is speculation not supported by data. I would remove this or rephrase it to a more scientific statement. Avoid using "important", "great impact".
  11. Paragraph 3.11/line 473: specify which experimental results you refer to and why.
  12. Most figures are mismatched and not appropriately cross-referred (e.g., figure 1, 3,  6, 7, 8? etc.)  Others are missing in the papers (e.g., figure 4D). On other occasions, the caption was not matching with the figure contents. Therefore, a full and conclusive paper evaluation was impaired. 
  13. Check the English and fix minor errors.

Round 2

Reviewer 2 Report

No other comments.